# High-Throughput Phenotyping Accelerates the Dissection of the Phenotypic Variation and Genetic Architecture of Shank Vascular Bundles in Maize (*Zea mays* L.)

**DOI:** 10.3390/plants11101339

**Published:** 2022-05-18

**Authors:** Shangjing Guo, Guoliang Zhou, Jinglu Wang, Xianju Lu, Huan Zhao, Minggang Zhang, Xinyu Guo, Ying Zhang

**Affiliations:** 1College of Agronomy, Liaocheng University, Liaocheng 252059, China; guoshangjing@lcu.edu.cn (S.G.); zhouguoliang96@sina.com (G.Z.); 2Beijing Key Lab of Digital Plant, Research Center of Information Technology, Beijing Academy of Agriculture and Forestry Sciences, Beijing 100097, China; wangjl@nercita.org.cn (J.W.); luxj@nercita.org.cn (X.L.); zh3321621294@163.com (H.Z.); zhangmg@nercita.org.cn (M.Z.)

**Keywords:** maize shank, micro-CT, vascular bundle, phenotypic variation, genome-wide association studies

## Abstract

The vascular bundle of the shank is an important ‘flow’ organ for transforming maize biological yield to grain yield, and its microscopic phenotypic characteristics and genetic analysis are of great significance for promoting the breeding of new varieties with high yield and good quality. In this study, shank CT images were obtained using the standard process for stem micro-CT data acquisition at resolutions up to 13.5 μm. Moreover, five categories and 36 phenotypic traits of the shank including related to the cross-section, epidermis zone, periphery zone, inner zone and vascular bundle were analyzed through an automatic CT image process pipeline based on the functional zones. Next, we analyzed the phenotypic variations in vascular bundles at the base of the shank among a group of 202 inbred lines based on comprehensive phenotypic information for two environments. It was found that the number of vascular bundles in the inner zone (IZ_VB_N) and the area of the inner zone (IZ_A) varied the most among the different subgroups. Combined with genome-wide association studies (GWAS), 806 significant single nucleotide polymorphisms (SNPs) were identified, and 1245 unique candidate genes for 30 key traits were detected, including the total area of vascular bundles (VB_A), the total number of vascular bundles (VB_N), the density of the vascular bundles (VB_D), etc. These candidate genes encode proteins involved in lignin, cellulose synthesis, transcription factors, material transportation and plant development. The results presented here will improve the understanding of the phenotypic traits of maize shank and provide an important phenotypic basis for high-throughput identification of vascular bundle functional genes of maize shank and promoting the breeding of new varieties with high yield and good quality.

## 1. Introduction

Maize is highly adaptable and widely planted, which is of great significance for meeting global food demand and ensuring world food security [1]. The yield of maize is dependent on a strong source-sink relationship, and abundant source-sink and efficient flow are the best conditions. The shank, formed by the differentiation and development of the ear, is the branch stem connecting stem and ear and an important developmental step that connects ‘source’ to ‘sink’. The shank structure is similar to that of the stem, but it has its own characteristics. The shape, length, and thickness of each internode of the shank are different from those of the stem. For example, the internode length of the shank shortens gradually from base to top, and the diameter of each internode of maize shank usually increases from the base to the top [2,3]. Vascular bundles of the shank are responsible for transporting water and nutrients from stem to ear, determining their transport efficiency [4]. Phenotypic traits of the vascular bundle are closely related to material transportation and distribution of grain, which play an essential role in the high yield of maize [5,6,7]. In 2007, He et al. found that the number and area of shank vascular bundles in large-ear type maize were higher than those in small-ear type maize and that treatment with a growth regulator could significantly increase the number and area of shank vascular bundles, thus significantly increasing grain volume and weight [8]. Wang analyzed the relationship between shank length and corn yield and found that there was a certain correlation between them. Studies have shown that the number, size and distribution of shank vascular bundles directly affected the transport efficiency of assimilates from the source to kernels, eventually limiting crop yield [9,10].

The phenotypic characterization of vascular bundle traits reported in previous studies was still mainly in the stage of manual detection. However, manual detection of vascular bundle traits was strenuous and time-consuming work. What is more, many characteristics of important biological significance were unable to be detected by the manual test [11]. In the past five years, image-based stem phenotyping analysis methods and software have significantly developed to realize the quantitative analysis of vascular bundles phenotypic traits [11,12,13,14,15]. The robust and automated techniques of phenotyping vascular bundle traits will provide researchers with the ability to analyse vascular bundles at larger scales and higher precision than ever.

In recent years, with the rapid development of high-density single nucleotide polymorphism (SNP) genotyping and next-generation sequencing (NGS), genome-wide association studies (GWAS) have become a powerful tool for analyzing the genetic basis of the quantitative variations in complex crop traits [16,17]. For maize, since the publication of the B73 reference genome [18], many genes for important agronomic traits have been identified by GWAS, such as plant height [19,20,21,22], flowering time [23], ear height [21,22] and grain size [24,25]. Meanwhile, phenotype-genotype association analysis is also an important tool for revealing the phenotypic diversity and genetic structure of maize vascular bundles [9,10,11,26]. Based on advanced microscopic imaging technology and intelligent phenotyping analysis algorithms, the combination of abundant phenotypic and genotypic data will provide a breakthrough for understanding the genetic basis of vascular bundle formation and development in the shank.

In this study, micro-CT scanning technology was used to obtain the micro images of shanks of inbred maize lines at resolutions that could reach 13.5 um. The phenotypic traits of shank and vascular bundles could be automatically detected using the automatic vascular bundle phenotyping pipeline. The five categories of thirty-six extracted phenotypic traits included traits related to cross-section, epidermis zone, periphery zone, inner zone, and vascular bundle. Based on abundant phenotypic indicators of shank and vascular bundles, we analysed the phenotypic variations in shank vascular bundles among a group of 202 inbred lines and further conducted GWAS to reveal the genetic architecture of the shank vascular bundles. The results presented here will provide a new perspective for high-throughput identification of vascular bundle functional genes of maize shank and promote breeding of new varieties with high yield and good quality.

## 2. Methods

### 2.1. Material and Sample Collection

The total of 202 inbred maize lines used in this study belong to the maize natural population described by Yang et al., 2011 [27]. According to the Q-matrix of population structure, the 202 inbred maize lines were divided into four subgroups: Stiff stalk (SS) with 25 lines, non-stiff stalk (NSS) with 67 lines, tropical-subtropical (TST) with 51 lines, and an admixed group with 59 lines. The 202 inbred maize lines were planted in an experimental fields of the Beijing Academy of Agriculture and Foresting Sciences in Beijing (116.2783° E, 39.9416° N) and Sanya (109.1870° E, 18.3905° N). In Beijing, sowing took place at the experimental field of the Beijing Academy of Agriculture and Foresting Sciences on 9 June 2020. Each inbred line was planted in four-row plots with eight plants in each row. Each row was 2.1 m long, and there was 60 cm between rows. The same experimental materials were planted at the experimental base of the Beijing Academy of Agriculture and Foresting Sciences at Sanya in 2021. Sowing took place on 20 March 2021. Each inbred line was planted in two-row plots with eight plants in each row. Each row was 2.1 m long, there was 60 cm between rows. The base internodes of the shank of three duplicates for each inbred line were collected at the silking period (73 days after sowing) and soaked in an FAA solution (90:5:5 *v*/*v*/*v*, 70% ethanol:100% formaldehyde:100% acetic acid) immediately for later study.

### 2.2. CT Image Acquisition and Phenotypic Analysis of Maize Shank

In order to ensure the reliability and consistency of image collection, the standardized process of maize vascular bundle microscopic data collection independently constructed by our research group was adopted for CT image acquisition and phenotypic analysis of maize shank [11]. The process consists of 4 parts: (1) Sample preparation. First, 0.5–1.0 cm segments of the internodes at the base of the shank were dehydrated in a series of ethanol gradients as follows: 70% alcohol 1 d→100% alcohol 1 d →100% alcohol 1 d; next, samples were transferred to tertiary butyl alcohol, as follows: tertiary butyl alcohol 2 d →tertiary butyl alcohol 2 d; then samples were frozen at −80 °C for 2 d, and frozen samples were freeze-dried for 5 h in batches; finally, the dried samples were stained with solid iodine for later CT scanning. (2) Micro-CT scanning. Samples were scanned using the Skyscan 1172 X-ray computed tomography system (Bruker, Nazareth, Belgium) at 40 kV/250 mA. The distance between the X-ray source and the detector was 345.59 mm, the distance between the X-ray source and the sample was 259.85 mm, the scanning resolution was 13.55 µm, the scanning mode was 2 K mode (2000 × 1332 pixels), and the scanning time was 22 min. (3) CT image reconstruction. The raw image data reconstructed by Skyscan NRecon software (Bruker, Nazareth, Belgium), and a series of 8-bit image files (BMP) were obtained to extract and analyze phenotypic features. (4) Phenotyping analysis for CT image of maize shank. Here, based on a large number of previous CT images and manual annotation of the data set, we presented a deep learning-integrated phenotyping pipeline to automatically extract and quantify vascular bundles for different types of stem internodes [11]. The phenotyping pipeline consisted of three steps, i.e., detecting vascular bundles, identifying zones and phenotyping. To count the number of vascular bundles accurately, the initial candidate regions of vascular bundles were first obtained by the above semantic segmentation. Then, each candidate region had to be evaluated for whether it was a single valid vascular bundle. In a given CT image, the slicing zone of maize stem could be divided into three function-related zones, i.e., epidermis (EZ), periphery (PZ) and inner (IZ) zones. In the phenotyping step, we extracted traits of a maize stem and its vascular bundles from different perspectives, such as quantity, size, shape and distribution, and we calculated the corresponding characteristics of vascular bundles and function zones.

### 2.3. Phenotypic Data Analysis

Microsoft Excel 2013 was used to sort and analyze the experimental data from Beijing and Sanya and calculate for the two locations yield maximums, minimums, averages, standard deviations, etc. A correlation study using Spearman’s correlation as a distance metric was conducted for phenotypic traits. ANOVA and Duncan’s test were used to compare the phenotypic differences among different subgroups of maize shank in 202 inbred lines at *p* < 0.05 level.

Locations were regarded as environmental variables. The best linear unbiased prediction (BLUP) for the 36 phenotypic traits was estimated with the following linear mixed model in the lme4 package of the R software:*yi* = *μ* + *fi* + *ei* + *εi*
where *yi* is the BLUP value of individual *i*, *μ* is the grand mean for the two (Beijing and Sanya) environments, *fi* and *ei* are the ith genetic effect (inbred lines) and environment effect, respectively, and *εi* is the random error. Additionally, *μ* was considered to be a fixed effect, whereas *fi* and *ei* were random effects.

### 2.4. Heritability Analysis

Heritability refers to the percentage of genetic variation ($V_{A}$) that accounts for the total variation in the phenotype, generally denoted by $H^{2}$. It can be used to compare the relationships between genetic and environmental factors for a specific phenotypic variation. Heritability ($H^{2}$) was calculated for each trait as follows:$$H^{2}=\frac{V_{g}}{V_{g}+\frac{V_{GL}}{L}+\frac{V_{e}}{L*R}}\;$$
where *L* is the number of locations, *R* is the number of replications, and *V_g_*, *V_GL_* and *V_e_* represent, respectively, the genotypic variance, interaction between inbred lines and environment variance, and the error variance, respectively. The above analysis was performed in ASReml-R v.3.0 using the ‘asreml’ function of R package asreml [28].

### 2.5. Genome-Wide Association Study

Genotype data were obtained from the laboratory of Professor Jianbing Yan of Huazhong Agricultural University (download URL: www.maizego.org/Resources.html, accessed on 17 August 2021). After the genotype data were obtained for 202 inbred lines, PLINK was used to calculate the minimum allele frequency (MAF), the MAF of each SNP was counted, quality control was conducted, SNPs with MAF < 0.05 and call rate <0.9 were removed, and the number of remaining effective SNPs was 797058. PopLDdecay (version 3.41) [29] was used to calculate the linkage disequilibrium decay. Population structure was estimated using Admixture (version 1.3.0) based on the 202 inbred maize lines [30]. The total of 797,058 SNPs were used to calculate the relative kinship using TASSEL 5 [19]. Statistical associations between the 36 phenotypic traits and the genotypes were examined using GWAS’s multi-locus random SNP effect mixed linear model tool (R package “mrMLM” version 4.0; Wuhan, China) [31]. Six methods (mrMLM, FASTmrMLM, FASTmrEMMA, ISIS EM BLASSO, pLARmEB and pKWmEB) included in the function ‘mrMLM’ were used in our study. The *p* value was set at 1.25 ×10^−6^ (*p* ≤ 1/N, where N is the total number of SNPS in the whole genome). The default *p* value of 0.0002 was then used as the filter threshold to determine the significance of the SNPs associated with a given trait. The intersection SNPs obtained by all methods were regarded as significantly related to phenotypic traits and were considered to be more reliable results. All candidate genes are based on EnsemblPlants (http://plants.ensembl.org/Zea_mays/Info/Index, accessed on 10 January 2022) in the maize B73 reference genome (B73RefGen_v4) and the NCBI gene database (https://www.ncbi.nlm.nih.gov/gene, accessed on 17 January 2022) interpretation. Separately, the significant interactions between the genes and their related phenotypic traits were visualized using Cytoscape v3.7.2 (National Institute of General Medical Sciences, Bethesda, MD, USA).

## 3. Results

### 3.1. Phenotypic Analysis of Shank and Vascular Bundle

Shank, an important developmental step that connects ‘source’ to ‘sink’, is the branch stem connecting stem and ear in maize. The structure of the shank is similar to that of the stem, but it has its own characteristics. High-quality microscopic images of the shank can be obtained through micro-CT scanning at resolutions that reach 13.5 µm. In studying the CT scanning images, it was not difficult to find that the shapes of the shanks are significantly different from those of the stem. In addition to being nearly round (Figure 1A1,A2), the shape of the shank cross-section can be divided into several other shapes, including crescent shape (Figure 1B1,B2), bell shape (Figure 1C1,C2), horseshoe shape (Figure 1D1,D2), and irregular shape (Figure 1E1,E2). Vascular bundle phenotyping pipeline-vesselparser was used to achieve the automatic, high-throughput analysis of vascular bundle phenotypes of maize shank. The segmentation, identification and classification of different tissue areas, namely, the epidermis zone, periphery zone and inner zone, which corresponded to the anatomy of the shank, namely, the epidermis, periderm and pith, could be reasonably realized. Once the different functional zones were segmented, vascular bundles in each zone were extracted, and their phenotypic traits were detected automatically (Figure 1B2-1–B2-4). Based on the above phenotyping pipeline, 36 phenotypic traits of the shank were extracted at one time. According to the anatomical characteristics of the shank tissue, the 36 phenotypic traits can be divided into five categories: cross-section related (7 items), epidermis zone related (2 items), periphery zone related (7 items), inner zone related (7 items), and vascular bundle related (12 items) (Table 1).

To explore the relationships between the phenotypic traits of the shank, correlation analysis was carried out of the 36 phenotypic traits (Figure 2). There were significant positive correlations between the geometric and morphological characteristics of the shank cross-sections (such as SZ_SA, SZ_A, SZ_CA, SZ_CCA, SZ_P, and SZ_LA) and the geometric and morphological characteristics of the inner zone (IZ_A and IZ_T). The number and area of total vascular bundles in a cross-section (VB_N, VB_A) were positively correlated with the number and area of vascular bundles in the functional zone (IZ_VB_N, IZ_VB_A, and PZ_VB_N). However, there was a significant negative correlation between mean vascular bundle size-related traits (VB_SAAVE, VB_LAAVE, VB_CCAAVE, VB_PAVE, VB_AAVE, and VB_CAAVE) and density properties of a vascular bundle (PZ_VB_D, IZ_VB_D, and VB_D). The results showed that the larger the vascular bundle area, the smaller the vascular bundle density within the cross section of the shank. Moreover, different from stem basal internodes, the morphological and geometric characteristics of vascular bundles in the inner zone might be more important in the shank.

Hierarchical cluster analysis was performed to explore the relationships between the 36 phenotypic traits using Spearman’s correlation as a distance metric. The 36 accessions were clustered into four major groups, and the resulting dendrogram is shown in Figure 2. Group I mainly reflected the distribution properties of the vascular bundles, containing PZ_VB_D, IZ_VB_D, and VB_D. Group II was composed of the 10 phenotypic parameters representing the mean characteristics of vascular bundle shape and size. Geometric and morphological characteristics of the shank cross-section and inner zone and the number and area of the vascular bundles in the cross-sections and inner zones were gathered into group Ⅲ, and the remaining 10 phenotypic parameters, mainly representing epidermis zone-related and periphery zone-related traits, were gathered into group Ⅳ. The systematic and abundant indicators of maize shank vascular bundles provided the phenotypic data basis for the follow-up omics research.

### 3.2. Phenotypic Variations of Shank Vascular Bundles among a Group of 202 Inbred Lines

Based on the comprehensive phenotypic data, we analysed the phenotypic variations in the vascular bundles at the base of the shank among a group of 202 inbred lines. The frequency distribution of the 36 phenotypic traits in the 202 inbred lines showed a continuous variation, indicating that the micro-phenotypes of shank were typical quantitative traits controlled by polygenes. Wide phenotypic variations in vascular bundle size, morphology, number, distribution density and other characteristics in the cross-section and functional zones (epidermis, periphery, inner) were observed. The interesting thing was that the area of the inner zone (IZ_A), ranging from 64.688 mm^2^ to 214.114 mm^2^ with an average of 108.220 mm^2^, had the highest maximum variation of 3.310-fold in 202 inbred lines, followed by the total area of the vascular bundles in the inner zone (IZ_VB_A, 3.033–fold), the convex area of vascular bundles in the inner zone (IZ_VB_CA, 3.031–fold), the shank cross-section area traits (SZ_CCA, 2.748-fold; SZ_A, 2.695–fold), the thickness of the inner zone (IZ_T, 2.634-fold), and vascular bundle area traits in the inner zone (IZ_VB_A, 2.336-fold; VB_CAave, 2.120–fold) (Table 1).

Moreover, we analysed the changes in the vascular bundle traits at the base node of the shanks in the different subgroups of the 202 inbred lines (TST, NSS, SS and Mixed) (Table 2). Most, 22, of the 36 phenotypic traits showed significant differences among different subpopulations (*p* ≤ 0.05), mainly cross-section-related, inner zone-related and vascular bundle-related traits. For the cross-section-related traits, six phenotypic indicators showed significant differences among different subpopulations: cross-section area (SZ_A), convex area of the cross-section (SZ_CA), circumcircle area of the cross-section (SZ_CCA), short axis length of the cross-section (SZ_SA), perimeter of the cross-section (SZ_P), and long axis length of the cross-section (SZ_LA). Three inner-zone-related phenotypic traits had significant differences among subpopulations, including thickness of the inner zone (IZ_T), area of the inner zone (IZ_A), and density of the vascular bundles in the inner zone (IZ_VB_D). Nine phenotypic traits related to vascular bundles were significantly different among subpopulations: total area of vascular bundles (VB_A), density of vascular bundles (VB_D), average short axis length of vascular bundles (VB_SAave), average perimeter of vascular bundles (VB_Pave), average long axis length of vascular bundles (VB_LAave), average circumcircle area of vascular bundles (VB_ CCAave), average convex area of vascular bundles (VB_CAave), and average area of vascular bundles (VB_Aave). Five phenotypic traits related to periphery zone were significantly different among subpopulations: density of vascular bundles in the periphery zone (PZ_VB_D), convex area of vascular bundles in the periphery zone (PZ_VB_CA), total area of vascular bundles in the periphery zone (PZ_VB_A), thickness of the periphery zone (PZ_T), and area of the periphery zone (PZ_A). There were no significant differences in epidermis zone-related and periphery zone-related phenotypic traits among the four subgroups of the 202 inbred lines. In addition, Duncan’s test showed that the shank cross-section areas in the NSS subgroup were significantly larger than those in the SS, Mixed and TST subgroups, and the inner zone area (IZ_A) and inner zone thickness (IZ_T) in the NSS subgroup were much larger. Meanwhile, the total number of vascular bundles (VB_N) and the number of vascular bundles in the inner zone (IZ_VB_N) in the NSS subgroup were greater than those numbers in the SS, Mixed and TST subgroups, and the total area of the vascular bundles (VB_A) in the NSS subgroup was much higher than that in other subgroups. On the contrary, vascular bundle density (VB_D), vascular bundle density in the inner zone (IZ_VB_D), and vascular bundle density in the periphery zone (PZ_VB_D) in the SS, Mixed and TST subgroups were significantly higher than those in the NSS subgroup.

### 3.3. Heritability Analysis

Here, the heritability of the 36 phenotypic traits of maize shank and vascular bundles based on the data from two environments was calculated. The 36 phenotypic traits showed different heritability patterns, ranging from 0.036 to 0.711 (Figure 3). Nearly all items, 30 (accounting for 83% of all traits), had heritability >0.3, indicating that variability in shank micro-phenotypes was governed in large part by genetic factors. Based on the clustering analysis and heritability values, we selected 30 phenotypic indicators with heritability higher than 0.3, allowing for the identification of SNPs controlling their expression in revealing genetic regulation mechanisms.

### 3.4. Genome-Wide Association Analysis of Shank Vascular Bundles

#### 3.4.1. SNP Loci Localization and Candidate Gene Identifying

In this study, genome-wide association analyses of 30 phenotypic traits of maize shank and vascular bundles were conducted using the multi-locus random-SNP-effect mixed linear models in R package ‘mrMLM’ (version 4.0). A total of 806 significant associated SNPs (*p*-value < 1.3 × 10^−6^) were identified for the 30 traits, and 1245 unique candidate genes were annotated according to the maize B73 reference genome (B73 RefGen_v4). In order to reduce the false positives of GWAS and ensure the accuracy of candidate genes, only the SNPs loci and genes co-located by various methods were used in subsequent gene selection. Consequently, 186 significant SNPs were filtered, and 360 unique candidate genes were annotated. Among these genes, 199 genes were unique candidate genes only related to a single trait (Table 3). For these 199 unique candidate genes, function annotation was conducted through the NCBI Gene database, and 130 genes with more detailed functional annotation were obtained (Supplemental Appendix A). These candidate genes mainly encode proteins involved in lignin, cellulose synthesis, transcription factors, material transportation and plant development.

#### 3.4.2. Pathways Enriched by Functional Enrichment Analysis

Pathways enriched by functional enrichment analysis were performed on 130 unique candidate genes with detailed annotation. After uploading the candidate gene IDs of all 30 phenotypic traits to KOBAS 3.0, a total of 20 KEGG pathways (*p*-value < 0.05) were enriched. In addition, a total of 60 GO terms (*p*-value < 0.05) were enriched after uploading the candidate gene IDs to PlantRegMap. Among the five categories of the shank phenotypic traits, the candidate genes associated with shank cross-section-related traits were only enriched in 1 KEGG pathway, monoterpene biosynthesis (zma00902, *p* = 0.0174), and there were 17 GO terms, among which ‘Peroxidase activity’ (GO: 0004601, *p* = 0.00013) and ‘Antioxidant activity’ (GO: 0016209, *p* = 0.00036) had the highest significance. A total of 5 KEGG pathways were enriched in candidate genes for the periphery-zone-related traits (*p* < 0.05), mainly including ‘Biosynthesis of secondary metabolites’ (zma01110, *p* = 0.03799) and ‘Metabolic pathways’ (zma01100, *p* = 0.03423). A total of 16 GO items, among which ‘Peroxidase activity’ (GO: 0004601, *p* = 0.00011) and ‘Antioxidant activity’ (GO: 0016209, *p* = 0.00013) had the highest significance. A total of 10 KEGG pathways were enriched in candidate genes for the inner-zone-related traits (*p* < 0.05), mainly including ‘Biosynthesis of secondary metabolites’ (zma01110, *p* = 0.02601), ‘ABC transporters’ (zma02010, *p* = 0.01071), and ‘Steroid biosynthesis’ (zma00100, *p* = 0.02197). Among the 29 GO items, ‘Catalytic activity’ (GO: 0003824, *p* = 0.03986) and ‘Integral component of membrane’ (GO: 0016021, *p* = 0.0401) had the highest significance. The candidate genes associated with vascular-bundle-related traits were enriched in 8 KEGG pathways (*p* < 0.05), mainly including ‘Biosynthesis of secondary metabolites’ (zma01110, *p* = 0.01824), ‘Metabolic pathway’ (zma01110, *p* = 0.03246), and ‘Brassinolactone biosynthesis’ (zma00905, *p* = 0.01546). Among 18 GO items, ‘Brassinosteroid homeostasis’ (GO: 001026, *p* = 0.0154) had the highest significance (Figure 4A).

Moreover, functional enrichment analysis was performed on 100 shared associated genes with detailed annotation. A total of 3 KEGG pathways (*p*-value < 0.05) were enriched, including alpha-linolenic acid metabolism (zma00592, *p* = 0.0012), Biosynthesis of secondary metabolites (zma01110, *p* = 0.0336), and Metabolic pathways (zma01100, *p* = 0.0360). 20 GO terms (*p*-value < 0.05) were enriched, and ’Transferase activity, transferring hexosyl groups’ (GO: 0016758, *p* = 0.0050) and ‘Acetylglucosaminyltransferase activity’ (GO: 0008375, *p* = 0.0077) had the highest significance (Figure 4B).

#### 3.4.3. Trait-Gene Network Visualization

The gene-phenotypic trait network was constructed of 29 phenotypic traits and their related genes (VB_CAR has no candidate genes, so it did not appear in this network diagram). Traits and genes are shown in different shapes and sizes. For the 29 large octagon nodes, the 11 green nodes represent vascular-bundle-related traits (VB_SAave, VB_Pave, VB_N, VB_LWR, VB_LAave, VB_D, VB_CCAave, VB_CAave, VB_Aave, VB_A, and ARIVB), the 6 blue nodes represent cross-section-related traits (SZ_SA, SZ_P, SZ_LA, SZ_CCA, SZ_CA, and SZ_A), the 4 yellow nodes represent periphery zone-related traits (PZ_VB_N, PZ_VB_CA, PZ_VB_D, and PZ_A), the 7 red nodes represent inner zone-related traits (IZ_VB_N, IZ_VB_D, IZ_VB_CAR, IZ_VB_CA, IZ_VB_A, IZ_T, and IZ_A), and the 1 brown node represents an epidermis zone-related trait (EZ_A). Genes are represented by the small round nodes, and different colours indicate different attributes. The light grey nodes are genes that only correlate with one specific trait, and deep purple to light purple in the centre (the darker the colour, the more traits are associated) indicates multi-trait shared genes. There were 100 shared genes among the traits within and between the five categories (Figure 5).

## 4. Discussion

### 4.1. Advances in Detecting Phenotypic Traits of Shank Vascular Bundle

In this study, shank CT images were obtained using the standard process for stem micro-CT data acquisition, and its resolution could reach 13.5 µm. Compared with traditional paraffin sectioning and hand-cutting sectioning, CT image acquisition efficiency is greatly improved, and the image quality is guaranteed [9,32,33]. Moreover, high-throughput and accurate detection of the micro-phenotypes of the shank were realized for the first time using the vascular bundle phenotyping analysis pipeline were extracted at one time, including cross-section-related, epidermis-zone-related, periphery-zone-related, inner-zone-related, and vascular bundle-related traits. In previous studies, micro-phenotypic traits of maize shank obtained by manual detection were very limited, and many characteristics of important biological significance could not be detected by the manual tests [11]. Based on ‘functional zone’ image segmentation, quantitative analysis of phenotypic traits of vascular bundles in different regions of shanks could be achieved, which was more novel and accurate than manually measuring. For example, phenotypic indicators such as periphery zone area and vascular bundle area in the periphery zone were difficult to detect using traditional methods. Accurate and in-depth analysis of phenotypic characteristics of shank and vascular bundles is of great significance in understanding the phenotypic diversity of maize shank and provides an important phenotypic basis for high-throughput identifying of the vascular bundle functional genes of maize [34].

### 4.2. Phenotypic Variation of Shank and Vascular Bundles

As the vertical axis of the maize plant, the stem is the hub connecting the above-ground organs and the underground parts, and it participates in the transportation and circulation of water and nutrients in the whole plant [35]. Shank, as the part of the ‘shoot system’, is the only channel connecting the stem and the ear; it determines the nutrient transport efficiency of the ear, thus affecting the material accumulation of grain. Similar to the stem structure, the shank is composed of nodes and internodes, and the anatomical structure of vascular bundles is the same as the stem. It is consistent with the results of Zhang’s research in that the vascular bundles in the periphery zone of the shank were denser than those in the inner zone, but the average area of vascular bundles in the inner region was larger than that in the periphery region [11]. According to the CT scanning images, it was not difficult to find that the shapes of the shanks changed significantly compared with the third internode. In addition to being nearly round, as with the third internode, the cross-section of the shank also presented bell, crescent and other shapes. The third internodes were larger than the shank, and the cross-section area, vascular bundle number, and periphery zone of the third internode were higher than those for the shanks. However, the area of the single vascular bundle at the base node of the shank was larger than that at the third internode. In addition, the most significant micro phenotypic trait of the third internodes in the maize natural population was the average area of circularity of the inner zone (IZCir), followed by the average area of vascular bundles in the inner zone (IZVBAvArea), the number of vascular bundles in the inner zone (IZVBNum), the area of the periphery zone (PZArea) and the area of the inner zone (IZArea). The most significant micro phenotypic trait of the shank in the 202 inbred lines was the area of the inner zone (IZ_A), followed by the total area of vascular bundles in the inner zone (IZ_VB_A), the convex area of vascular bundles in the inner zone (IZ_VB_CA), and the shank cross-section area traits (SZ_CCA, SZ_A). These results indicated that phenotypic characteristics of vascular bundles in the inner zone may play a greater role in the shank, which may be closely related to material transportation and yield formation.

Phenotypic variations of 36 phenotypic indicators were different among subgroups. In this study, 1.001- to 3.310-fold variations in shank micro phenotypes were detected in 202 inbred maize lines. There were 22 phenotypic indexes (*p* ≤ 0.05), including SZ_A, VB_A, IZ_VB_A, etc. In addition to the traditional microscopic character indicators, such as SZ_A, VB_N, and VB_D, several new phenotypic traits like IZ_A, IZ_VB_D, IZ_VB_A were significantly different among different subpopulations, and these new phenotypic indexes can be used as essential referenc to distinguish different genotypes. In addition, for the NSS subgroup, the total area of shank vascular bundles was larger, and the total number of vascular bundles was larger, but the density of vascular bundles was smaller. On the contrary, the densities of shank vascular bundles in the SS, Mixed and TST subgroup were larger. These results indicated that the smaller and denser distribution of shank vascular bundles shank in tropical and subtropical inbreeds might be related to climate adaptation.

### 4.3. Candidate Genes Analysis

In recent years, high-throughput crop phenotyping techniques have been developing rapidly and microscopic phenotypic studies are in the ascendant. Micro-phenotype is an integral part of crop phenomics, and micro traits play an important role in the accurate identification and function prediction of specific genes [34]. Genome-wide association studies (GWAS) have been widely applied in the genetic dissection of various agronomic traits of crops and have decoded the functions of a large number of unknown genes [36,37]. In this study, we identified 199 unique candidate genes for 30 shank micro-phenotypic traits and 100 shared associated genes that were common to these traits, which to the best of our knowledge are the most systematic and abundant phenotypic traits of maize shank. These candidate genes encode proteins involved in lignin, cellulose synthesis, transcription factors, material transportation and plant development.

The relationships among 36 phenotypic traits were well analysed by correlation analysis and cluster analysis, and the identified correlated traits could be explained by associated candidate genes that were common to such traits based on GWAS results. We discover that several phenotypes that share genes generally belong to the same functional region or the same traits. For example, vascular bundle average-size-related traits (such as VB_CCAave, VB_Aave, VB_Pave, VB_CAave, VB_SAave, and VB_LAave) generally share the same associated candidate genes. Cross-section size-related traits (SZ_A, SZ_CA, SZ_CCA, etc.) and inner zone size-related traits (IZ_A) share associated candidate genes, such as Zm00001d047436, Zm00001d047359, Zm00001d047358, Zm00001d037842, Zm00001d033954, and Zm00001d027392. The total vascular bundle number trait (VB_N) shares candidate genes with the trait of vascular bundle number in the periphery zone (PZ_VB_N), such as Zm00001d039323 and Zm00001d017462. Further, the vascular bundle density trait (VB_D) was negatively correlated with cross-section size-related and functional zone size-related traits, and these traits also had associated candidate genes, such as Zm00001d021936, Zm00001d020580, and Zm00001d002123. These shared genes provide useful information for finding functional links between correlated shank phenotypes.

A set of genes encodes proteins that are involved in lignin, cellulose synthesis and cell wall formation, which are associated with vascular bundle number and area traits. Candidate genes GRMZM2G029144 and GRMZM2G156257, which regulate the trait of number of vascular bundles in the periphery zone (PZ_VB_N), has been mapped to chromosome 2 (chr2.S_219749444) and encodes a peroxidase-related protein peroxidase 2-like [38]. Another candidate gene GRMZM2G033985, which regulates the trait of the total area of vascular bundles (VB_A), has been mapped to chromosome 10 (chr10.S_46801060) and encodes a peroxidase-related protein peroxidase 40. Peroxidase plays central roles in lignin formation and cell wall biosynthesis [39,40]. Three maize cDNAs coded for three different peroxidases (ZmPox1, ZmPox2 and ZmPox3) were isolated, and ZmPox2 and ZmPox3 were involved in lignification [41]. We speculate that expression of peroxidase may accelerate the cell wall or fibre biosynthesis process of vascular bundles in shank internodes. The finding also suggests that peroxidase may also be involved in regulating total vascular bundle area and vascular bundle number in the periphery zone. Moreover, the candidate gene GRMZM2G014560, which regulates the trait of vascular bundle size (VB_LWR), has been mapped to chromosome 5 (chr5.S_191129001) and is involved in wall-associated receptor kinase. It is a special plant receptor kinase covalently cross-linked with pectin in the cell wall [42]. There is also a family of WAK-like proteins (WAKL) similar to WAK in plant structuress, and WAK and WAKL may be important proteins in the connection and communication between plant cell wall and cytoplasm [43]. The candidate gene GRMZM2G002523, which regulates the density of vascular bundles in the inner zone (IZ_VB_D), was mapped to chromosome 2 (chr2.S_187823360) and involved in cellulose biosynthesis [44]. Many studies have shown that the biosynthesis of plant cellulose is a complex process of cellulose synthase complex (Rosette) composed of CesA and other enzymes [45,46]. This gene may be involved in cellulase synthesis and thus affect vascular bundle density traits.

A set of candidate genes is involved in vascular bundle and plant development. The candidate gene GRMZM2G000743, which regulates the trait of density of vascular bundles (VB_D), has been mapped to chromosome 1 (chr1.S_276307427) and is involved in encoding homeobox protein KNOX3. Recent discoveries indicate that KNOX genes promote xylem fibre differentiation during vascular development [47]. A study of loss-of-function mutations of rice OSH15 has demonstrated its role in the development of internodes [48]. These results strongly suggest that the homeobox KNOX genes play important roles in the differentiation of vegetative tissues. The significant SNP located on chromosome 1 at position 6194262, which regulates the trait of average convex area ratio of vascular bundles (VB_CCAAVE), is contained in the gene region of GRMZM2G180951 that encodes callose synthase. Callose synthesis is carried out by callose synthases (CalS), otherwise known as glucan synthase-like (GSL) [49]. In plants, callose deposits control the symplasmic exchange of essential nutrients and signalling macromolecules in plasmodesmata and phloem and thus control the distribution and development of plants [50,51]. For example, CalS7 is specifically found in the phloem and correlated with an increase in the deposition of callose in the phloem in Arabidopsis thaliana and tomatoes [52,53,54]. Another gene, GRMZM2G026742, which regulates the density of vascular bundles in the periphery zone (PZ_VB_D), w mapped to chromosome 9 (chr9.s_99531345), encoding Heat Shock Transcription Factor 9. Heat shock transcription factors (Hsfs) are the most important transcription regulators. Hsf genes have been identified in tomato, rice, A. thaliana, Zea mays, Cucumis sativa, Solanum tuberosum, and other plants [55,56,57,58,59]. In addition to heat stress, Hsfs are involved in plant growth and other biotic and abiotic stress responses [60,61]. It has found that HsfA9 was involved in embryo development and seed maturation in A. thaliana and Helianthus annuus [62].

Candidate genes related to MYB transcription factor family genes were jointly identified in the third internodes and the shanks. The candidate gene GRMZM2G172327, regulating the trait of area ratio of individual vascular bundles (ARIVB), was mapped to chromosome 7 (chr7.s_155367082), belonging to the MYB transcription factor family. This gene encodes transcription factor MYB21, which has been implicated in the development of flower vascular bundles in Arabidopsis thaliana [63]. In addition, studies have shown that MYB21 might be the regulator of CWIN gene expression, and cell wall invertase (CWIN) hydrolyses sucrose into glucose and fructose in the extracellular matrix and plays crucial roles in assimilate partitioning and sugar signaling [64,65]. MYB genes have shown that they influenced a cell’s shape and differentiation, and they were activated during hormone responses and plant defence reactions [66].

The phenotype-associated genes obtained in our study can provide a reference for related research and offer new ideas for exploring the genetic mechanisms of shank vascular bundle agronomic traits in future studies.

## Figures and Tables

**Figure 1 plants-11-01339-f001:**
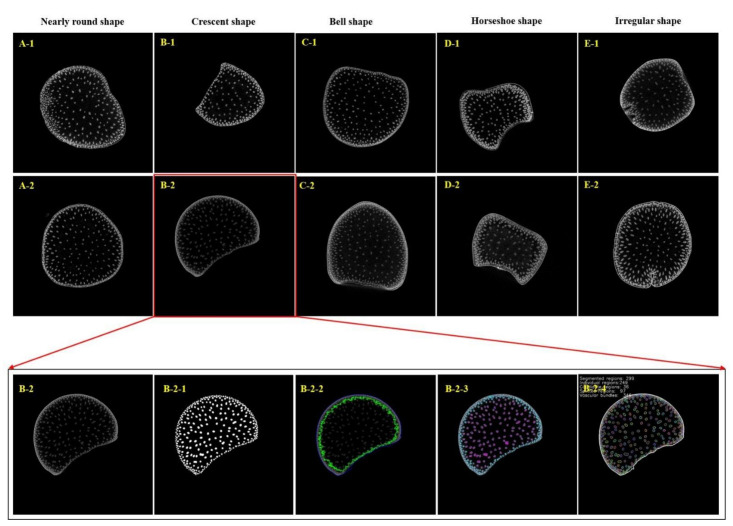
The shape variations of maize shank cross-sections. (**A1**,**A2**): Nearly round shape; (**B1**,**B2**) crescent shape; (**C1**,**C2**) bell shape; (**D1**,**D2**) horseshoe shape; (**E1**,**E2**) irregular shape). The image processing results for a maize shank cross-section. ((**B2**) Source image; (**B2-1**) vascular bundle segmentation result; (**B2-2**) the boundaries of the epidermis, periphery and inner zones (the blue line defines the boundary of the epidermis, and the green line segments the boundary between the periphery and inner zones); (**B2-3**) classification of vascular bundles in different functional zones (blue represents the vascular bundles in the periphery zone, and rose red represents the vascular bundles in the inner zone; (**B2-4**) vascular bundle labelling and phenotypic index calculation).

**Figure 2 plants-11-01339-f002:**
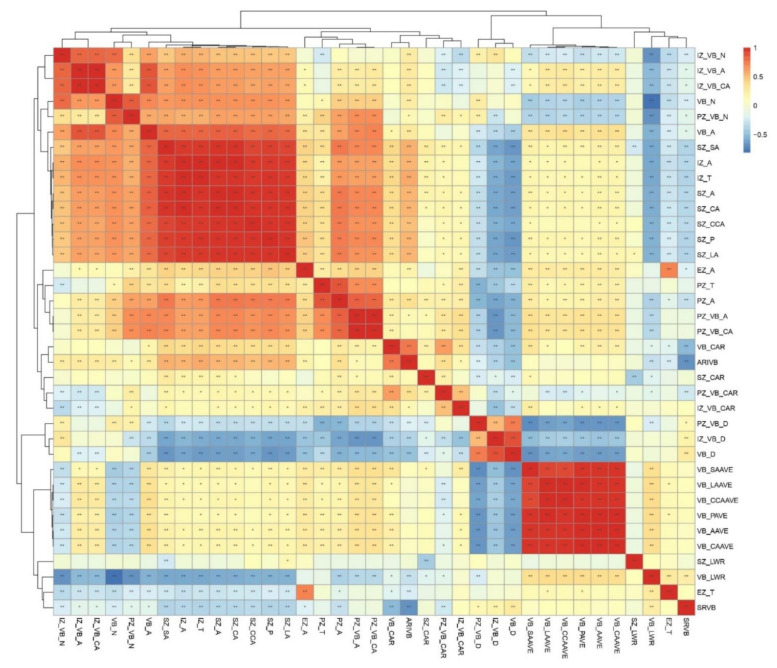
Spearman’s correlations of the 36 phenotypic traits of the shank basal internodes in natural maize populations in two environments. ** *p*-value ≤ 0.01, * *p*-value ≤ 0.05.

**Figure 3 plants-11-01339-f003:**
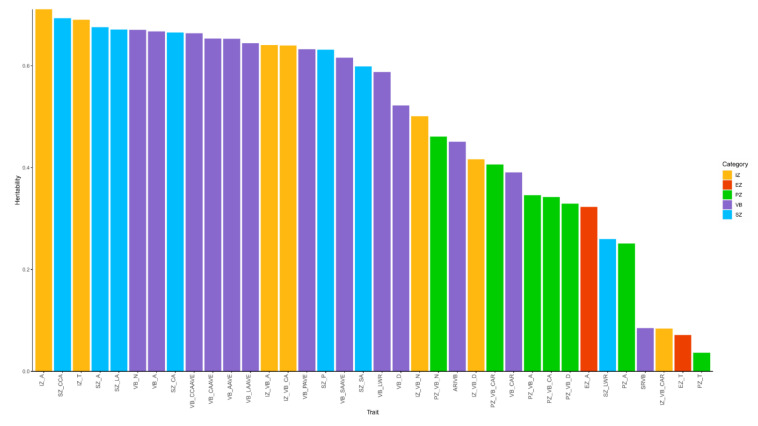
The heritability (*H*^2^) of the investigated 36 phenotypic traits of maize shank. Yellow columns represent inner zone-related traits (7 items), red columns represent epidermis zone-related traits (2 items), green columns represent periphery zone-related traits (7 items), purple columns represent vascular bundle-related traits (12 items), and blue columns represent cross-section-related traits (7 items).

**Figure 4 plants-11-01339-f004:**
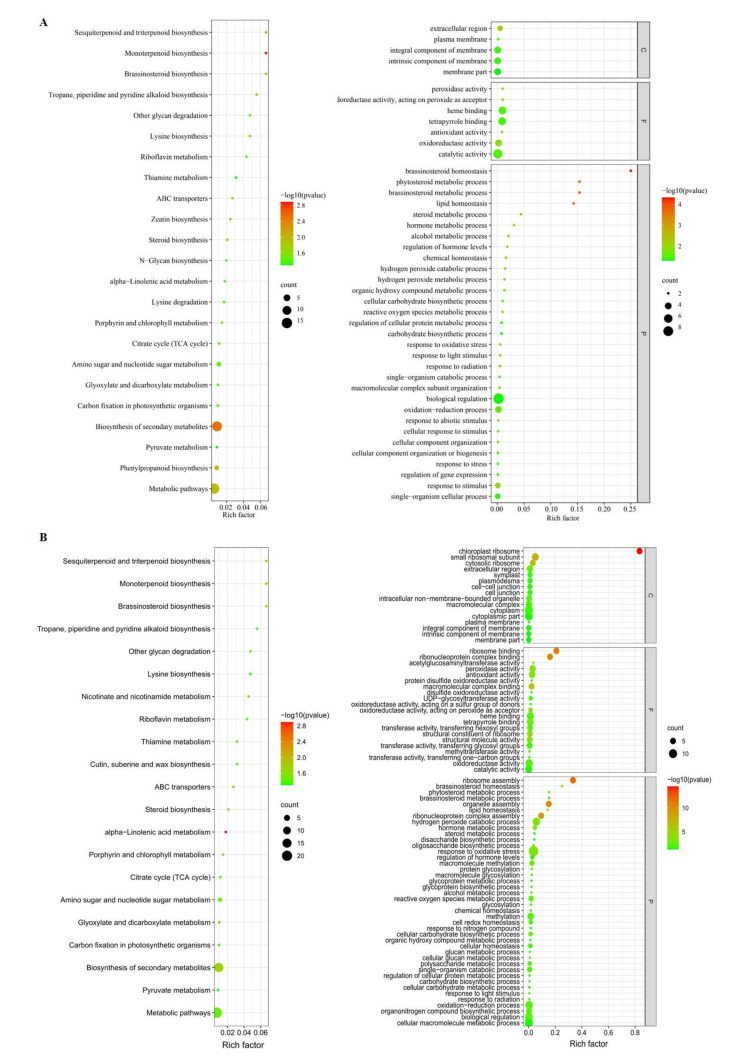
GO terms and KEGG pathways enriched by the genes associated with 36 phenotypic traits. (**A**) GO terms and KEGG pathways enriched by 130 unique candidate genes. (**B**) GO terms and KEGG pathways enriched by 100 shared associated genes. Left: KEGG pathways. Right: GO items. C represents cellular component, F represents molecular function, and P represents biological processes.

**Figure 5 plants-11-01339-f005:**
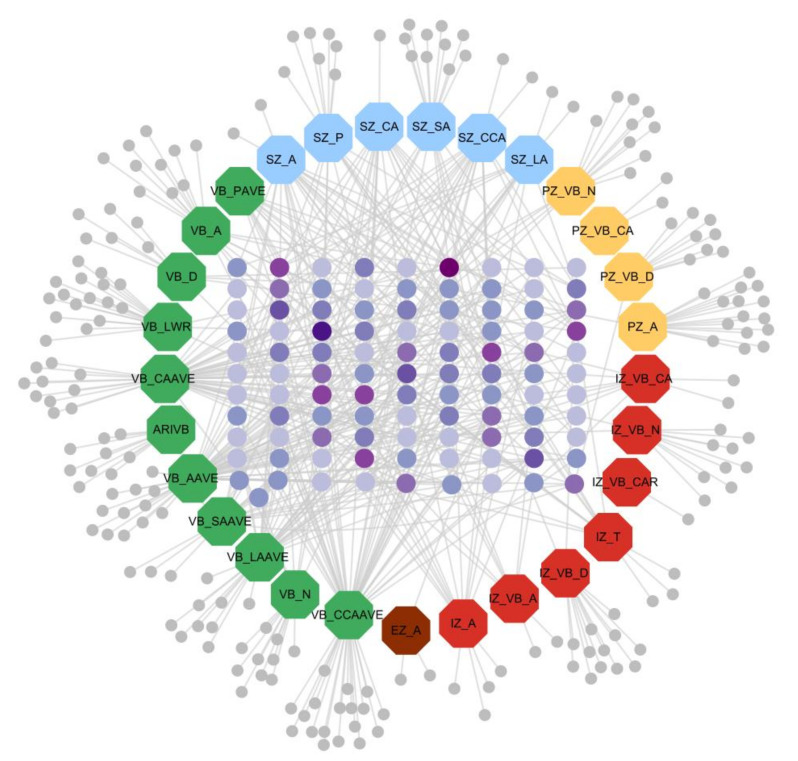
The gene-phenotypic trait network constructed by 30 phenotypic traits and their related genes. For the 29 large octagon nodes, the 11 green nodes represent vascular-bundle-related traits (VB_SAave, VB_Pave, VB_N, VB_LWR, VB_LAave, VB_D, VB_CCAave, VB_CAave, VB_Aave, VB_A, and ARIVB), the 6 blue nodes represent cross-section-related traits (SZ_SA, SZ_P, SZ_LA, SZ_CCA, SZ_CA, and SZ_A), the 4 yellow nodes represent periphery zone-related traits(PZ_VB_N, PZ_VB_CA, PZ_VB_D, and PZ_A), the 7 red nodes represent inner zone-related traits (IZ_VB_N, IZ_VB_D, IZ_VB_CAR, IZ_VB_CA, IZ_VB_A, IZ_T, and IZ_A), and the 1 brown node represents an epidermis-zone-related trait (EZ_A). Genes are represented by the small round nodes, and different colours indicate different attributes. The purple round nodes represent the overlapped genes of multiple traits, and there were 100 overlapped genes (purple round nodes represent the overlapped genes of 11 traits, indigo round nodes represent the overlapped genes of 8 traits, dark violet round nodes represent the overlapped genes of 7 traits, dark orchid round nodes represent the overlapped genes of 6 traits, blue violet round nodes represent the overlapped genes of 5 traits, medium orchid round nodes represent the overlapped genes of 4 traits, medium slate blue round nodes represent the overlapped genes of 3 traits, and medium purple round nodes represent the overlapped genes of 2 traits). The darker the colour, the more traits are associated; the light grey round nodes stand for genes only have correlation with specific traits.

**Table 1 plants-11-01339-t001:** The phenotypic variations in the vascular bundles at the base internode of the shank among a group of 202 inbred lines.

Group	Phenotypes	Index	Unit	Median	Minimum	Maximum
Cross-section	The short axis length of the slice zone	SZ_SA	mm	12.267	7.537	18.678
	The perimeter of the slice zone	SZ_P	mm	44.903	34.325	128.45
	The length-width ratio of the slice zone	SZ_LWR	-	1.098	1.008	1.547
	The long axis length of the slice zone	SZ_LA	mm	13.557	8.649	21.574
	The circumcircle area of the slice zone	SZ_CCA	mm^2^	132.604	55.483	318.124
	The convex area of the slice zone	SZ_CA	mm^2^	137.63	59.087	232.311
	The area of the slice zone	SZ_A	mm^2^	134.354	55.833	319.316
Epidermis zone	The thickness of the epidermis	EZ_T	mm	0.076	0.052	0.144
	The area of epidermis	EZ_A	mm^2^	3.344	1.5	5.876
Periphery zone	The number of vascular bundles in the periphery zone	PZ_VB_N	-	193	91	424
	The density of vascular bundles in the periphery zone	PZ_VB_D	number/mm^2^	7.861	3.181	15.198
	The average convex area ratio of the vascular bundles in the periphery zone	PZ_VB_CAR	-	0.944	0.482	0.962
	The convex area of the vascular bundles in the periphery zone	PZ_VB_CA	mm^2^	12.494	5.581	25.588
	The total area of the vascular bundles in the periphery zone	PZ_VB_A	mm^2^	11.731	5.311	24.114
	The thickness of the periphery zone	PZ_T	mm	0.687	0.291	1.209
	The area of the periphery zone	PZ_A	mm^2^	25.675	9.536	47.102
Inner zone	The number of vascular bundles in the inner zone	IZ_VB_N	-	154	67	338
	The density of the vascular bundles in the inner zone	IZ_VB_D	number/mm^2^	1.529	0.571	3.481
	The average convex area ratio of the vascular bundles in the inner zone	IZ_VB_CAR	-	0.976	0.964	0.98
	The convex area of the vascular bundles in the inner zone	IZ_VB_CA	mm^2^	15.281	6.725	35.693
	The total area of the vascular bundles in the inner zone	IZ_VB_A	mm^2^	14.912	6.561	34.76
	The thickness of the inner zone	IZ_T	mm	5.709	3.498	9.299
	The area of the inner zone	IZ_A	mm^2^	108.220	64.688	214.114
Vascular bundles	The average short axis length of the vascular bundles	VB_SAave	mm	0.274	0.222	0.395
	The average perimeter of the vascular bundles	VB_Pave	mm	1.036	0.82	1.466
	The total number of vascular bundles	VB_N	-	357	196	722.5
	The length-width ratio of vascular bundle	VB_LWR	-	0.004	0.002	0.008
	The average long axis length of the vascular bundles	VB_LAave	mm	0.33	0.257	0.47
	The density of the vascular bundles	VB_D	number/mm^2^	2.718	1.035	5.605
	The average circumcircle area of the vascular bundles	VB_CCAave	mm^2^	0.107	0.064	0.213
	The average convex area ratio of the vascular bundles	VB_CAR	-	0.966	0.953	0.975
	The average convex area of the vascular bundles	VB_CAave	mm^2^	0.08	0.051	0.158
	The average area of the vascular bundles	VB_Aave	mm^2^	0.077	0.049	0.154
	The total area of the vascular bundles	VB_A	mm^2^	27.16	13.587	56.158
	The separation ratio of the vascular bundles	SRVB	-	2.726	1.831	9.613
	The area ratio of individual vascular bundles	ARIVB	-	0.725	0.424	0.899

**Table 2 plants-11-01339-t002:** The 22 phenotypic traits of the shank base internodes that showed significant differences among the subpopulations of 202 inbred lines (TST, NSS, SS and Mixed).

Phenotypic Index	*p*-Value	Mixed	NSS	SS	TST
SZ_SA (mm^2^)	0.00016 ***	12.631 b	13.094 a	12.378 b	12.266 b
SZ_P (mm)	0.000841 ***	45.812 b	47.537 a	45.229 b	44.71 b
SZ_LA (mm)	0.000143 ***	13.918 b	14.577 a	13.806 b	13.524 b
SZ_CCA (mm^2^)	0.000274 ***	175.382 b	191.394 a	170.184 b	164 b
SZ_CA (mm^2^)	8.19 × 10^−5^ ***	145.432 b	159.054 a	140.723 b	137.129 b
SZ_A (mm^2^)	4.59× 10^−5^ ***	143.297 b	157.433 a	138.59 b	134.456 b
PZ_VB_D (number/mm^2^)	3.82× 10^−5^ ***	6.859 b	6.661 c	7.071 a	6.997 ab
PZ_VB_CA (mm^2^)	0.000825 ***	13.594 ab	13.86 a	13.173 b	13.16 b
PZ_VB_A (mm^2^)	0.00085 ***	12.72 ab	12.976 a	12.324 b	12.308 b
PZ_T (mm)	0.000152 ***	0.779 ab	0.781 a	0.776 c	0.777 bc
PZ_A (mm^2^)	4.06× 10^−6^ ***	30.92 b	32.089 a	29.936 b	30.04 b
IZ_VB_D (number/mm^2^)	0.00201 **	1.467 a	1.393 b	1.506 a	1.5 a
IZ_T (mm)	0.000684 ***	5.784 b	6.057 a	5.733 b	5.596 b
IZ_A (mm^2^)	0.000395 ***	108.716 b	120.298 a	106.58 b	101.691 b
VB_SAave (mm)	0.018 *	0.29 ab	0.292 a	0.283 b	0.285 b
VB_Pave (mm)	0.0268 *	1.095 ab	1.103 a	1.07 b	1.075 b
VB_LAave (mm)	0.0331 *	0.352 ab	0.354 a	0.343 b	0.345 b
VB_D (number/mm^2^)	0.000197 ***	2.549 a	2.409 b	2.621 a	2.62 a
VB_CCAave (mm^2^)	0.0267 *	0.122 ab	0.124 a	0.116 b	0.116 b
VB_CAave (mm^2^)	0.0171 *	0.089 ab	0.091 a	0.085 b	0.086 b
VB_Aave (mm^2^)	0.0156 *	0.086 ab	0.088 a	0.082 b	0.083 b
VB_A (mm^2^)	0.000822 ***	28.436 ab	29.686 a	27.568 b	26.689 b

Note: 0 ‘***’, 0.001 ‘**’, 0.01 ‘*’, 0.05; the same index with different lowercase letters indicates significant difference at 0.05 level.

**Table 3 plants-11-01339-t003:** Summary of significant loci from the genome-wide association study.

Trait	No. of Significant SNPs	No. of Annotated Genes	No. of Significant SNPs Validated by Multiple Methods	No. of Unique Annotated Genes Validated by Multiple Methods	No. of Genes Only Related to Specific Trait	No. of SNPs Shared between More than 2 Phenotypes	No. of Shared Associated Genes between More than 2 Phenotypes
SZ_A	32	61	7	14	2	19	21
SZ_P	29	54	9	15	7	11	11
SZ_LA	35	67	5	10	2	10	12
SZ_SA	36	67	8	16	10	9	9
SZ_CA	23	43	9	17	1	15	18
SZ_CCA	36	67	7	14	2	17	19
EZ_A	18	32	1	2	2	1	1
VB_N	34	54	9	18	10	6	6
VB_A	35	60	7	12	10	8	8
VB_AAVE	109	186	22	42	13	40	47
VB_PAVE	28	52	6	11	0	10	13
VB_LAAVE	53	92	18	33	9	19	26
VB_SAAVE	55	96	6	10	4	14	16
VB_CAAVE	96	163	21	37	10	36	42
VB_CCAAVE	127	211	21	40	18	32	40
VB_CAR	1	2	0	0	0	0	0
VB_LWR	50	70	12	21	16	3	3
PZ_VB_N	43	73	7	12	10	3	3
PZ_A	37	66	12	23	15	1	1
IZ_VB_N	33	55	6	11	9	2	2
IZ_T	26	40	9	13	3	10	10
IZ_VB_A	26	44	6	12	2	7	7
IZ_A	15	24	5	10	4	11	12
ARIVB	30	56	7	13	9	0	0
PZ_VB_D	27	45	6	9	8	1	1
PZ_VB_CA	32	50	6	12	2	0	0
IZ_VB_D	18	34	8	16	12	2	2
IZ_VB_CA	19	33	7	12	2	7	7
IZ_VB_CAR	4	7	1	1	1	0	0
VB_D	27	47	6	12	6	4	4
Summery	806	1245	186	320	199	88	100

## Data Availability

Genotypic data that support the findings of this research are open resource and can be downloaded from http://www.maizego.org/, accessed on 17 August 2021. All other data are available from corresponding author upon reasonable request.

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
