# Peer review of "High-Throughput Phenotyping Accelerates the Dissection of the Phenotypic Variation and Genetic Architecture of Shank Vascular Bundles in Maize (Zea mays L.)"

_plants, 2022, doi:10.3390/plants11101339_

Round 1
Reviewer 1 Report
Review of MS "High-Throughput Phenotyping Accelerates the Dissection of the Phenotypic Variation and Genetic Architecture of Shank Vascular 3 Bundles in Maize Natural Population" (plants-1652975) by Guo et al.
In this work the authors carried detailed morphological analyses of the maize shank and a GWAS of 30 shank phenotypic traits in a panel of 202 inbred lines. They used state of the art micro-X ray computer tomography techniques and developed an automated phenotyping pipeline to acquire all the data from images. Their work represents an interesting advancement to understand the natural variations in shank structure and development in maize, a very important organ that is at the core of the control of the source-sink relationship and influences grain yield. I praise the authors for the large amount of work and data obtained. However, I would like to see improvements in their manuscript in terms of the English language and the presentation and discussion of their results before it is accepted for publication.
Minor points:
Title: The term "natural population" does not apply to this study since the authors are dealing with a domesticated plant. I suggest to change the title to: "High-Throughput Phenotyping Accelerates the Dissection of the Phenotypic Variation and Genetic Architecture of Shank Vascular Bundles in Maize (Zea mays L.)".
The manuscript requires a strong revision of the English language. These are a just few examples for improvement:
- Abstract (lane 17): "In this study, the shank CT images...." Change to: "In this study, shank CT images...."
- Abstract (lane 22): "....natural population subgroup (202 inbred lines)...". Comment similar to point 1. Change to: ".... a group of 202 inbred lines...". Change throughout the manuscript all "natural population" phrases in a similar manner.
- Abstract (lanes 28-29): "These candidate genes encode mainly involved in lignin, cellulose synthesis, .....". Change to: "These candidate genes encode proteins involved in lignin and cellulose synthesis, ...."
- Abstract (lanes 31-32): "....for high-throughput identifying of vascular bundle...", Change to: "....for high-throughput identification of vascular bundle..."
- Introduction (lane 40): ".....closely related to the source-sink.....", change to: "...is dependent on a strong source-sink relationship...."
- Introduction (lane 42): "....and the last step from ‘source’ to ‘sink’..", change to: ".... an important developmental step that connects ‘source’ to ‘sink’.."
- Introduction (lanes 51-52): "In 2007, He et al found that....", change to: "In 2007, He et al. found that...."
- Lanes (58-59): "....transporting efficiency....", change to: "....transport efficiency...."
- Lanes (61-62): "Most phenotypic detections of the vascular bundle were still mainly in the stage of manual detection". Change to: "Phenotypic characterization of vascular bundle traits reported in previous studies was still mainly in the stage of manual detection".
- Lane 62: " However, manually detection of ...". Change to: " However, manual detection of ..."
- Lane 65: "...methods and soft have significantly...", change to: "...methods and software have significantly...".
- (Many more corrections are needed between lanes 65 and 88).
- Lane 88: "......shanks of maize natural population....", change to: "......shanks of maize inbred lines...."
Major points:
- Materials and Methods section (lanes 101-110): The authors do not describe locations, growth seasons nor years of experimental plantings of the inbred lines under study. This is important as they claim to have measured heritability and environmental effects.
- Which methods were used for genomic sequencing and SNP detection?
- Show the linkage disequilibrium decay plot for SNP data
- Results section. Which maize inbred line(s) was(were) used in experiments described under section 3.1?
- Lane 290: Describe each one of the mixed linear models used for GWAS contained in the R package ‘mrMLM’ (version 4.0).
- Show QQ plots for each GWAS method used and add them to Supplementary materials section.
- Provide Manhattan Plots for each GWAS method used and add them to Supplementary materials.
- There is no detailed description or discussion of the strong positive or negative correlations observed between the phenotypic traits shown in Figure 2. What do these patterns mean in terms of shank development and/or function?
- Lanes 274-275: To which two environments the authors refer to? I could not find anywhere in the ms a description of this. Section 2.1. only describes one environment.
- The strong correlations (positive or negative) found between the 30 phenotypic traits analyzed has some influence in the identity of the candidate genes shared between different traits (Table 3)? If this is the case, can you discuss the role of such shared candidate genes in those phenotypic characters?
- The authors should be consistent with the nomenclature of maize genes in their ms. They mapped SNPs in version 4.0 of the B73 reference genome. However, in Results section they used previous versions of maize genes (V2 and V3), while in Supplementary information they used V 4.0.
- The authors made a poor job in describing and discussing their data, they could also discuss it more in depth. For example, they mentioned the importance of peroxidases in lignification (lanes 401-404), but failed to identify GRMZM2G033985 (lane 400) (Zm00001d024119) as such or identify other peroxidases among the 130 candidate genes found in Supplementary information.
- There are many candidate genes with interesting functions that are not mentioned at all and are relevant to their work. For example, KNOX3 gene (Zm00001d033861, Supplementary Information), encoding a homeobox protein that known to regulate plant development is not mentioned or discussed at all.
- In another example, the authors talk about a candidate gene encoding a Heat Shock Transcription Factor [lane 411] (GRMZM2G026742 or Zm00001d048041, encoding Heat Shock Transcription Factor 9) limiting their discussion to its obvious function given by its name (stress-related) and missing the fact that many HSFs also control plant development.
- Lane 420: Feng et al., 1934 should be corrected to: Feng et al., 2011.
- If several of their candidate genes are cell wall related, I wonder why the authors did not try to carry simple cell wall comparative histochemical analyses in shank tissues of inbred lines containing minor versus major alleles associated to such candidate genes?
- The authors also are encouraged to discuss their results in studying shank vascular bundles in maize with a paper published by them last year (Zhang et al., 2021) on vascular bundles of the third internode, alsi in maize. What are the differences/similarities in morphology and candidate genes associated to the traits analyzed in both studies?
Author Response
Dear Reviewer,
Thank you very much for your reviewers’ comments concerning our manuscript entitled “High-throughput phenotyping accelerates the dissection of the phenotypic variation and genetic architecture of shank vascular bundles in maize natural population” (plants-1652975). Those comments are very valuable and helpful for revising and improving our paper, as well as the important guiding significance to our research. We have studied comments carefully and have made a comprehensive correction which we hope meet with approval. We are looking forward to hearing from you regarding our submission and we will be glad to respond to any further questions and comments that you may have.
Revised portions are marked in blue in the paper. The main corrections in the paper and the response to the reviewer’s comments are as an attachment file. Please refer to the attachment for details.

Reviewer 2 Report
The authors report phenotyping 36 traits of maize shanks, their variations among 4 subgroups (SS, NSS, TST, Mixed) of 202 inbred lines, and many SNPs associated with these traits.
2.2. Although the authors mentioned phenotyping pipelines with machine learning in Introduction, the authors did not provide the source and citation of the pipeline used in this study. Readers are difficult to access and verify this method.
2.3. How did you treat phenotypic values from the two experimental fields to calculate the statistics? The descriptions seems copies from the application manuals. In BLUP, genetic effects are inbred lines? environmental effects are two experimental fields? Explain them in your study design.
2.4. Why did not you use Admixture software for so many SNPs instead of STRUCTURE? How did you incorporate genetic structure and environmental factors into mrMLM? (and I cannot find STRUCTURE outputs in Results)
3.1. Show trait groups (3 major groups and IV) in Figure 2.
3.2. These results are complicated and difficult to understand. Relationship between the 5 trait categories in Table 1 and the 4 trait groups (I – IV) from clustering are unclear.
3.3. The 4 trait groups (I – IV) or the 5 trait categories should be displayed (different colors) in Figure 3.
4. Most discussions are repeats of results. Add some interpretations and implications such as L384–386.
Author Response

(The authors gave the same response as above.)

Round 2
Reviewer 1 Report
I am pleased about the way the manuscript is improving after the first round of revision. Derived from one of my earlier suggestions (to describe more carefully the strong correlation, positive or negative, among different phenotypes), I included a few additional major points for revision. At the end of this second review, I enlisted some new minor points to be considered for revision.
Major points:
- Correlation analyses in Fig. 2 identified positively- and negatively-correlated traits that could be explained by associated SNPs that are common to such traits. I do not see any reason to exclude the collection of shared associated-genes from the Results and Discussion sections. Actually, these genes could be very useful to find functional links between such correlated phenotypes. Thus, please include, in Suppl. Information, SNPs and associated genes shared between phenotypes, not only those that are unique. In addition, in Table 3 include a new column with the number of associated SNPs that were shared between more than 2 phenotypes and a second new column with the number of the associated genes shared between more than 2 phenotypes (are they represented by 100 associated genes, as I can gather from data in Figure 5?).
- Expand the functional enrichment analysis (section 3.4.2) with the group of shared associated-genes. Likewise, in Figure 4 include the GO term and KEGG pathway information for the group of shared associated-genes.
- In Figure 5 legend (lane 381), please explain: " The purple round nodes represent the overlapped genes of multiple traits ...). How many overlapped genes did you find? Please state the number of overlapped genes.
- Discuss the possible functional relations of the shared genes with the phenotypes that they associate with (section 4.3).
- Scientific research is about reproducibility. Therefore, I encourage the authors to include, as Supplemental information, the following:
- All data pertaining to the phenotypic values for each one of the 220 inbred lines.
- GWAS performed using each one of the 6 different methods described in section 5.
Readers will be benefited greatly by the inclusion of this supplementary information. These data will be useful to perform, in the future, comparisons and/or meta-analyses.
Minor points:
- Title: (.... Zea Mays). Species name (mays), must be written with lowercase letters. Please make correction.
- Lanes 228-230 (2nd version). Improve the syntax, please.
- Figure 1 Legend. Explain which colors define the epidermis, periphery, and inner zone of the shank in panel B-2-2 and the vascular bundles in B-2-3.
- Table 1 and Table 2 (pages 8 and 9). What are the units of value for the phenotypes shown in these Tables? Please clarify.
- Discussion section: (lane 411): The following sentence is not quite clear: " Shank, as a branch of the stem, is the abnormal stem." . Please clarify in terms of its developmental biology.
- lane 414: "........the shank is the horizontal axis of the maize plant.." Actually, branches extend from the main axis at different angles, why do you consider the shank to be only on the horizontal plane?
- lane 445: This statement is confusing: ".... which new phenotypic indexes.." Perhaps change it to: "....such new phenotypic indexes.."?
- lane 448: Change : "... vascular bundles was more," to: ".... vascular bundles was larger,".
Author Response
Dear Reviewer,
Thank you very much for your reviewers’ comments concerning our manuscript entitled “High-Throughput Phenotyping Accelerates the Dissection of the Phenotypic Variation and Genetic Architecture of Shank Vascular Bundles in Maize (Zea mays L.)” (plants-1652975). According to your latest questions and comments, we have made a comprehensive correction which we hope to meet with approval.
Revised portions are marked in green in the paper. The main corrections in the paper and the response to the reviewer’s comments are as the attachment for details.

Round 3
Reviewer 1 Report
The new version of the manuscript is much better. However, upon revision, I found one major and a few minor items that need correction:
Major:
- Table S2: This table is missing some important information.
- Columns H, I, J, and N lack headings.
- The "Map location column" (M) is empty. Please fill in the missing data.
- The "OMIM" column (T) is empty. Please fill-in the missing data. In addition, explain the meaning of this abbreviation, it is not defined.
- The locations of the 88 SNPs that map near the genes described in columns C, H, and I are not identified. Authors should do something similar to what was done in columns I and J in Table S1.
MInor:
- Line 21: "...among a group of 202 inbred lines..." Please revise the number of inbred lines studied. According to the Abstract (above), and to Table S3, the study was made with 202 inbred lines. This is in conflict with line 559: ".....for 30-item phenotypic traits of the 220 inbred lines...."
- Figure 4B. The "Rich factor" phrase is missing on the x-axis of each one of the two columns.
- Line 426: The sentence: "Shank, as the part of the ‘flow', is ....". I suggest: ""Shank, as the part of the ‘shoot system´, is ....".
- Line 557: The sentence: "100 sheared associated can- ". Change to: "100 shared associated can- "
Author Response
Dear Reviewer,
Thank you very much for your reviewers’ comments concerning our manuscript entitled “High-Throughput Phenotyping Accelerates the Dissection of the Phenotypic Variation and Genetic Architecture of Shank Vascular Bundles in Maize (Zea mays L.)” (plants-1652975). According to your latest questions and comments, we have made correction which we hope to meet with approval. Thank you again for your careful review of our manuscript.
